# Interpretable Variational Autoencoder with Stabilized Tree Regularization

## Abstract

With the growth of digital health, bioinformatics and healthcare now produce massive high-dimensional (HD) datasets that challenge both prediction and interpretability. This work introduces the Tree-Regularized Interpretable Variational Autoencoder (TRI-VAE), which couples a VAE with a surrogate decision tree to impose rule-consistent structure on the latent space. TRI-VAE aligns embeddings to soft leaf distributions for a cluster-aware representation learning, and employs a SHAP-based attribution scheme tailored to HD settings to select salient features and harmonize feature-level explanations with path-based rules. A tree-regularizer, optimized via a learned average-path-length surrogate, promotes compact and balanced trees; stability-controlled tree updates further preserve assignment consistency over training. Across public (TCGA-LIHC, TUEP) and private (PPH) datasets, TRI-VAE delivers competitive predictive performance while yielding faithful, human-readable explanations. An LLM-assisted evaluation protocol with clinician review supports the accessibility and reliability of the extracted rules and attributions, advancing trustworthy AI for medical data analysis.

## 1 Introduction

In the rapidly evolving field of digital health, data-driven innovations are reshaping the medical landscape. Advanced medical devices, gene sequencing technologies, and widespread electronic health record systems generate massive amounts of high-dimensional (HD) data (Kasoju et al., 2023), unlocking significant potential for smart healthcare. These rich datasets enable accurate prediction and interpretation of health data patterns, benefiting the healthcare industry by supporting precise diagnoses and personalized treatment for practitioners, early disease detection and prevention for patients, and accelerated research progress for medical researchers (Yeung et al., 2023). However, fully leveraging complex HD data for accurate predictions and understanding the underlying decision-making processes remains a major challenge.

Explainable prediction with HD data involves two key issues: (i) reducing dimensionality to enable accurate prediction and (ii) providing an interpretability mechanism that ensures the comprehensibility and credibility of the results. This work employs a Variational Autoencoder (VAE) to learn structure for dimensionality reduction, and combines post-hoc SHAP attributions with tree-regularization–based intrinsic interpretability to analyze the model's decision process, enhancing reliability and transparency in practice. Integrating these three components introduces several challenges: first, VAE has inherently complex and obscure internal mechanisms, and existing methods struggle to guide its learning process effectively from an interpretability standpoint. Second, standard SHAP estimators can be computationally inefficient and high-variance in high dimensions. Lastly, the interpretability of the tree model becomes less intuitive as the number of nodes in the decision tree increases, making the decision-making process harder to understand.

To address these issues, this work introduces the Tree-Regularized Interpretable Variational Autoencoder (TRI-VAE) for high-dimensional data. TRI-VAE incorporates an explanatory guidance mechanism that guide VAE training with a surrogate decision tree, aligning latent variables to rule-consistent soft leaf distributions and improving representation quality. The framework further introduces a SHAP estimation scheme tailored to high-dimensional settings to quantify feature contributions, and integrates the resulting attributions with the tree's intrinsic structure. This work presents a new interpretability mechanism that integrates the intrinsic interpretability of tree regularization

with the post-hoc interpretability of SHAP, offering an intuitive and balanced explanation of the model's decision-making process.

TRI-VAE is validated through comprehensive experiments on public and domain-specific private datasets, benchmarked against strong baselines with component-wise evaluations to assess robustness. An automated LLM-based protocol is further introduced to evaluate interpretability, providing both theoretical and practical support for real-world application.

The contributions of this work are summarized as follows:

- This work proposes a theoretical framework of the Tree-Regularized Interpretable Variational autoencoder (TRI-VAE) to address the interpretability challenges in high-dimensional predictive modeling (§ 4.5).

- An explanatory mechanism is introduced to guide VAE's learning process, strengthening the connection between the latent representations learned by VAE and the decision tree structure, thus enhancing model interpretability (§ 4.2).

- A more efficient algorithm is developed for computing SHAP values, reducing computational costs and improving feature selection, thereby enhancing model training and analysis efficiency (§ 4.3).

- An integrated interpretability mechanism is introduced that combines intrinsic interpretability with post-hoc interpretability, contributing toward the goal of trustworthy AI (§ 4.4).

- To assess interpretability, this paper uses LLM to generate questions, which are validated by a medical professional to ensure the validity and trustworthiness of the model(§ 5.6).

## 2 RELATED WORK

**Variational autoencoders and variants.** VAEs (Kingma, 2013) widely used to learn low-dimensional structure from high-dimensional data (Mattei & Frellsen, 2019). Adding a classifier on top of the learned embeddings often yields strong downstream classification performance on complex data(Zhang et al., 2019; Hira et al., 2021). However, these methods often lack interpretability, which is particularly concerning in medical settings. To mitigate this issue, XOmiVAE incorporates DeepSHAP to attribute importance to input features and latent dimensions in omics-based cancer classification(Withnell et al., 2021; Lundberg & Lee, 2017). Yet these approaches remain post-hoc and do not induce a rule-consistent decision structure in the input space, leaving the internal workings of VAEs largely opaque. In contrast, this work couples a VAE with an interpretable, tree-guided module that aligns latent representations to soft leaf distributions, yielding rule-consistent partitions in latent space and traceable decisions.

**SHAP and Shapley Value.** SHAP is a widely used post-hoc interpretability framework that quantifies feature importance for individual samples, providing insights into model predictions. Exact Shapley values are computationally prohibitive due to factorial-scale permutations. To address this, two main approximation methods have been developed: model-specific methods, exemplified by TreeSHAP (Lundberg et al., 2018), which exploit tree structure to compute Shapley values without sampling (Yu et al., 2022; Muschalik et al., 2024; Yang, 2021); and model-agnostic methods such as Kernel SHAP (Lundberg & Lee, 2017), which fit a local, kernel-weighted linear surrogate and derive Shapley values on that approximation. The key advantage of this approach is its independence from any specific model structure, which makes it highly versatile and has been shown to be effective in various tasks (Chau et al., 2022; Covert & Lee, 2021; Aas et al., 2021). However, Kernel SHAP's sampling estimator becomes costly and high-variance in high dimensions. This study refines Kernel SHAP with subset-size–aware coalition sampling consistent with the Shapley kernel, improving sample efficiency and stability while preserving its theoretical basis.

**Decision Trees and Tree regularization.** Decision trees (Loh, 2011) are standard supervised-learning models for classification and regression. Their hierarchical structure yields transparent, human-simulatable rules that are easy to visualize and scrutinize, making trees natural vehicles for interpretability (Arrieta et al., 2020; Burkart & Huber, 2021). A growing body of work has sought to advance the interpretability of decision trees. Souza et al. (2022) optimize split structure to minimize

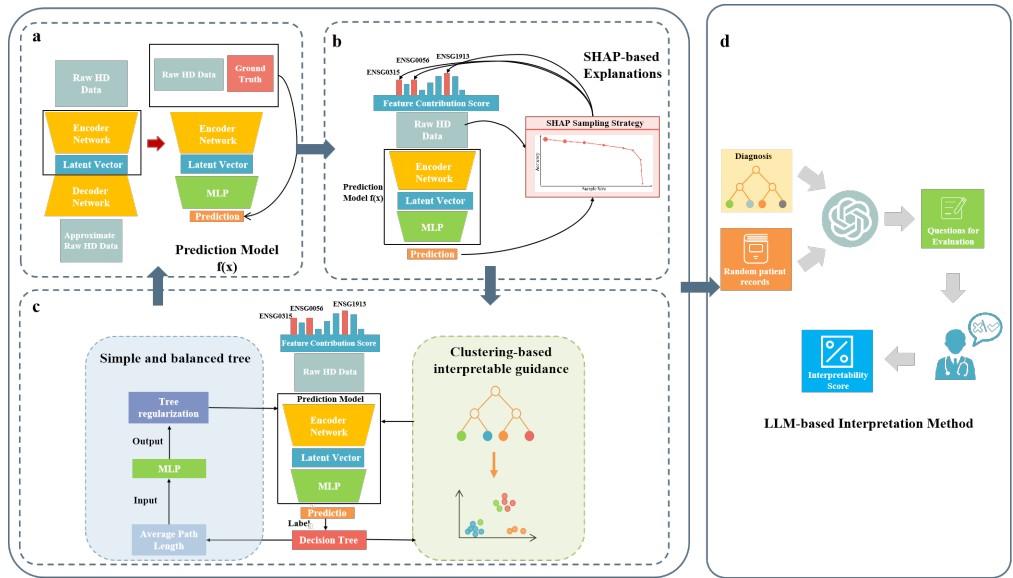

Figure 1: The framework of Tree-Regularized Interpretable Variational Autoencoder.

explanation size, yielding shorter, simpler rules without sacrificing accuracy; Wang et al. (2023) employ trees to extract disentangled factors from neural policies and propose evaluation metrics that operationalize interpretability in robotics; and Arenas et al. (2022) formalize explanation notions and derive complexity bounds that delineate the feasibility of exact extraction. Yet purely tree-based methods still struggle to form semantically meaningful splits in high dimensions. To bridge expressive deep models with interpretable rules, Frosst & Hinton (2017) distill deep networks into soft decision trees, achieving competitive accuracy while exposing explicit, path-based rules; Wu et al. (2018) regularize deep networks with a surrogate decision tree by penalizing its average path length (APL) as a regularization term, so the deep model remains well-approximated by a compact, human-simulatable tree. Leveraging this regularization, this study applies a learned APL surrogate to penalize surrogate-tree complexity and promote compact, balanced trees, and employs a stability-controlled tree-update protocol to reduce update-induced drift.

## 3    PRELIMINARIES

High-dimensional data suffer from the "curse of dimensionality", and many deep learning models operate as "black boxes", complicating interpretability. To address this, XOmiVAE combines a VAE with SHAP. This approach uncovers the contributions and correlations of genes and latent dimensions in cancer classification, enabling novel biomedical insights. In XOmiVAE, the VAE performs dimensionality reduction by learning an encoder $q_\phi(z \mid x)$ and a decoder $p_\theta(x \mid z)$, mapping high-dimensional data $x \in \mathbb{R}^d$ to a lower-dimensional latent space $z$. The training objective is the evidence lower bound (ELBO):

$$\mathcal{L}_{\text{VAE}} = \mathbb{E}_{z \sim q_\phi(z|x)}[\log p_\theta(x \mid z)] - D_{\text{KL}}(q_\phi(z \mid x) \,\|\, p_\theta(z)). \tag{1}$$

A prediction head then computes a soft label $\hat{y}(x)$ for each data point, which is used for attribution.

Let $N = \{1, \ldots, d\}$ index features and let $f(x)$ denote the scalar output to be explained. For any subset $S \subseteq N$, define the characteristic function

$$v(S \mid x) = \mathbb{E}_{X_{N \setminus S}}\Big[f\big(x_S, X_{N \setminus S}\big)\Big]. \tag{2}$$

where the expected model output when the features in $S$ are fixed to their observed values $x_S$ and the remaining features are integrated under a background distribution. The Shapley value of feature $i \in N$ is

$$\phi_i(x) = \sum_{S \subseteq N \setminus \{i\}} \frac{|S|! \, \big(d - |S| - 1\big)!}{d!} \Big(v(S \cup \{i\} \mid x) - v(S \mid x)\Big). \tag{3}$$

which quantifies the marginal contribution of feature $i$ across coalitions. XOmiVAE thus reduces dimensionality while improving interpretability by using SHAP values $\{\phi_i(x)\}$ to explain feature contributions to predictions.

# 4 METHODOLOGY

## 4.1 PROBLEM FORMULATION

Given $\mathcal{D} = \{(x_i, y_i)\}_{i=1}^N$ with $x_i \in \mathbb{R}^d$ and $y_i \in \{0, 1\}$, the goal is to learn a mapping $f_\theta : \mathbb{R}^d \to \{0, 1\}$ that combines predictive accuracy and interpretability. As in Fig. 1, TRI-VAE combines (i) a VAE for compact latent representations ($\mathcal{L}_{\text{VAE}}$), (ii) an efficient SHAP-based attribution module for feature selection, (iii) tree-guided clustering to align embeddings with rule-based partitions ($\mathcal{L}_{\text{cluster}}$), and (iv) a tree regularizer that penalizes average path length to promote concise, balanced tree ($\mathcal{L}_{\text{tree}}$). Sections 4.2–4.4 detail these components.

## 4.2 INTERPRETIVE GUIDANCE ARCHITECTURES

Although VAEs can learn rich latent distributions, they offer limited insight into the model's internal mechanisms. The interpretive guidance architecture addresses this by introducing structural supervision into the VAE's latent space through a surrogate decision tree. This tree defines an interpretable partition, with each leaf corresponding to a rule defined by its root-to-leaf path. A soft label is assigned to each sample to capture its compatibility with all leaf rules. The network then aligns its predictions with these soft labels, encouraging samples with similar leaf-rule profiles to cluster in the representation space.

**Tree-guided soft labels and alignment.** Given a dataset $\mathcal{D}$, a surrogate decision tree $T$ evaluates each sample $x_i$ through a series of splits, producing leaf–rule scores that induce a soft distribution $p(x_i)$ over all leaves. Note that computations are restricted to an attribution-selected feature subset to avoid performance degradation in high-dimensional settings (see §4.3). Let $L = \{\ell_1, \ell_2, \ldots, \ell_m\}$ be the set of unique leaves observed across samples, where $m = |L|$. Each leaf $\ell_j$ is assigned a unique cluster index $c \in \{1, \ldots, m\}$. The soft-label matrix $\mathbf{Y}_{\text{soft}} \in \mathbb{R}^{N \times m}$ is then defined by $[\mathbf{Y}_{\text{soft}}]_{i,c} = p_c(x_i)$.

To construct $p(x)$, let the root–to–leaf constraint sequence for leaf $\ell_j$ be $\mathcal{L}_j = \{(f_k, \theta_k, b_k)\}_{k=1}^{r_j}$, where $f_k$ is the split feature, $\theta_k$ the threshold, and $b_k \in \{+1, -1\}$ encodes the branch orientation of $\ell_j$ at node $k$. To measure the difference between leaves, the distance between the path and the sample $d_j$ is calculated, which quantifies how well a sample satisfies the corresponding path constraints. For a sample x, node-wise violations accumulate only when a split condition is not satisfied. Formally, the violation at each split is defined as

$$d_j(x) = \frac{1}{r_j} \sum_{k=1}^{r_j} \max\bigl(0, b_k(x_{f_k} - \theta_k)\bigr). \tag{4}$$

With leaf $\ell_j$ assigned to cluster $c$, soft labels are then obtained by normalizing the path scores into a distribution over clusters.

$$[\mathbf{Y}_{\text{soft}}]_{i,c} = p_c(x_i) = \frac{\exp\bigl(-d_j(x_i)\bigr)}{\sum_{u=1}^m \exp\bigl(-d_u(x_i)\bigr)}. \tag{5}$$

The network is then required to predict, for each sample, a distribution over the same $m$ clusters. Concretely, a linear mapping from the latent representation produces logits $h(x_i) \in \mathbb{R}^m$ and $\hat{y}(x_i) = \text{softmax}\bigl(h(x_i)\bigr)$. Define $\hat{y}_c(x_i) = [\hat{y}(x_i)]_c$, where $\hat{y}_c(x_i)$ denote the predicted probability that sample $x_i$ belongs to cluster $c$. The alignment is enforced with the Kullback–Leibler (KL) divergence:

$$\mathcal{L}_{\text{cluster}} = \frac{1}{N} \sum_{i=1}^N \sum_{c=1}^m [\mathbf{Y}_{\text{soft}}]_{i,c} \log \frac{[\mathbf{Y}_{\text{soft}}]_{i,c}}{\hat{y}_c(x_i)}. \tag{6}$$

Minimizing this clustering loss aligns representations with the tree-induced partition of the input space, making samples that follow similar decision paths closer in the latent space. This alignment

enhances the synergy between the neural network and the surrogate tree, yielding more interpretable guidance.

**Cluster-Head Update and Inheritance** Under soft supervision, training proceeds in a tree-guided manner: at iteration $t$ the network is distilled into a surrogate tree $T_t$ whose leaves induce clusters and soft labels. When the tree is replaced by the updated tree $T_t^{\text{new}}$, the clustering head is first resized to match the new leaf count and then updated via a structure-aware transfer to preserve cluster identities from $T_t^{\text{prev}}$. Specifically, each new leaf is paired with the most compatible predecessor leaf using two complementary criteria—*path similarity*, which measures agreement of root-to-leaf split sequences and *sample overlap*, computed as the intersection-over-union(IoU) between assignments induced by $T_t^{\text{prev}}$ and $T_t^{\text{new}}$. If both criteria indicate a reliable match, the corresponding output weights and bias are initialized by convex combination between the default initializer and the parameters of the matched predecessor (see Appendix B.1, Algorithm 1 for implementation details); otherwise, the default initializer is used and unmatched predecessors are discarded. This procedure yields smooth cluster evolution under minor edits while remaining robust to substantive changes, maintaining consistent soft labels and a stable optimization trajectory.

**Controlled Tree Update** To mitigate update-induced drift that destabilizes cluster identities, TRI-VAE performs tree replacement as a controlled update with separate scoring and gating stages. Let $T_t^{\text{cur}}$ be the in-force tree and $T_t^{\text{cand}}$ the tree distilled at iteration $t$ on an attribution-selected feature subset $U_t$ with attribution scores $\boldsymbol{s}_t$ (see §4.3). The candidate is first evaluated by a penalized fidelity objective that balances predictive agreement with structural simplicity:

$$J(T_t^{\text{cand}}) \;=\; \text{MSE}(T_t^{\text{cand}}) \;+\; \lambda_{\text{apl}} \cdot \text{APL}\big(T_t^{\text{cand}}; X_{\text{all}}\big), \tag{7}$$

where MSE measures fidelity to the network's outputs, and APL denotes the average path length of the surrogate tree (formally defined in §4.4), computed over the entire input set $X_{\text{all}} = \{x_1, \ldots, x_N\}$ to penalize excessive depth.

Acceptance of a new tree is then gated by three complementary criteria: (i) *attribution alignment*, requiring sufficient correlation between split importance and attribution scores ($\text{align}(T_t^{\text{cand}}, \mathbf{s}_t) \geq \tau_{\text{align}}$), ensuring consistency with feature-level explanations; (ii) *assignment stability*, quantified by the normalized mutual information $\text{NMI}(\mathbf{z}_t^{cur}, \mathbf{z}_t^{cand})$ between successive leaf assignments (where $\mathbf{z}_t^{cur}$ and $\mathbf{z}_t^{\text{cand}}$ are the vectors of per-sample leaf indices produced by $T_t^{\text{cur}}$ and $T_t^{\text{cand}}$, respectively) , which is permutation-invariant and robust to differing leaf counts; and (iii) *fidelity improvement*, requiring that the penalized objective $J$ strictly decreases. Metric definitions and implementation details are provided in Appendix C and Appendix B.2 respectively.

### 4.3 Efficient SHAP Sampling Strategies

Training surrogate trees in high-dimensional spaces can yield deep, high-variance structures. TRI-VAE therefore restricts the surrogate to a compact, attribution-selected feature subset ranked by SHAP that reflect the network's predictive behavior and align naturally with the tree splits. Instead of standard Kernel SHAP – whose kernel weighting suffers from degeneracy and ill-conditioning as dimensionality grows, this method rebalances the sampling budget across subset sizes while preserving Shapley-consistent weights, achieving broad coalition coverage at comparable cost.

**Subset-size–weighted sampling.** Specifically, weights are assigned to all possible subset sizes. Let $d$ denote the number of input features, $U \subseteq \{1, \ldots, d\}$ a feature subset, and $\rho_{|U|}$ the total weight assigned to all subsets of size $|U|$. This weighting enables effective sampling across subset sizes.

$$\rho_{|U|} = \frac{d-1}{|U|(d-|U|)}. \tag{8}$$

Subsequently, this sampling approach, based on Kernel SHAP, generates feature subsets such that the number of sampled subsets of a given size is proportional to the total weight assigned to that subset size. Let $k$ be the total number of sampled subsets. The number of samples assigned to subsets of a specific size $|U|$, denoted by $k_{|U|}$, is determined according to the following equation:

$$k_{|U|} = k \times \frac{\rho_{|U|}}{\sum_{i=1}^{d-1} \rho_i}. \tag{9}$$

Interestingly, for very small and very large subset sizes, it is possible to generate more samples than the number of possible subsets, which could provide deeper insights into these extreme cases.

Finally, to compute the weight for each individual sample, the total subset size weight is distributed among the subsets. This is accomplished using the following equation:

$$\pi_{|U|} = \frac{\rho_{|U|}}{k_{|U|}}. \tag{10}$$

**Attribution smoothing and feature selection.** To stabilize feature ranking across epochs, an exponential moving average (EMA) is maintained over attribution estimates. Let $\varphi^{(t)} \in \mathbb{R}^d_{\geq 0}$ be the kernel–weighted, magnitude–averaged attribution vector aggregated at epoch $t$. The smoothed scores are

$$s^{(t)} = \beta\, s^{(t-1)} + (1-\beta)\, \varphi^{(t)}, \qquad \beta \in (0,1),\ s^{(0)} = \varphi^{(0)}. \tag{11}$$

In practice, $\beta$ was empirically chosen as $0.6$. At each epoch, the top–$\kappa$ features under $s^{(t)}$ form the feature subset $U_t$, with $\kappa = \min\big(\lceil 0.25\, d \rceil,\ \kappa_{\max}\big)$, and the surrogate tree is then trained on $U_t$.

### 4.4 SIMPLE AND BALANCED DECISION TREE MODELS

To keep the surrogate decision tree both simple and balanced, a tree regularizer (Wu et al., 2018) is adopted. Minimizing the regularizer $L_{\text{tree}}$ promotes the formation of compact and well-balanced trees. Here, the complexity is quantified by the average path length (APL), defined as the expected number of internal decisions along the root-to-leaf path required to make a prediction for a given sample $x$.

Because tree induction is non-differentiable, the regularizer cannot be optimized directly. A small MLP surrogate $\hat{\Omega}(W; \xi)$ is therefore trained to map the current network parameters $W$ to an estimate of the tree's APL. Let $\Omega(W)$ denote the measured APL of the surrogate tree fitted at parameters $W$. During training, parameter snapshots and their measured complexities form a dataset $\mathcal{S} = \{(W_j, \Omega(W_j))\}_{j=1}^J$, and the surrogate is learned by $L_2$-regularized least squares:

$$\min_{\xi} \sum_{j=1}^{J} \big(\Omega(W_j) - \hat{\Omega}(W_j; \xi)\big)^2 + \varepsilon \|\xi\|_2^2. \tag{12}$$

The tree regularizer used in optimization is then taken to be the surrogate output,

$$L_{\text{tree}} \equiv \hat{\Omega}(W; \xi). \tag{13}$$

To remain aligned with the evolving model, only parameter–APL pairs from the most recent 50 epochs are retained when updating the surrogate.

### 4.5 TREE-REGULARIZED INTERPRETABLE VARIATIONAL AUTOENCODER

TRI-VAE optimizes a weighted sum of three objectives: (i) a VAE term that reconstructs high-dimensional inputs and shapes a compact latent space; (ii) a cluster-alignment term that matches network predictions to tree-induced soft labels; and (iii) a tree-complexity term that penalizes excessive average path length to promote simple and balanced rules.

$$\mathcal{L} = \mathcal{L}_{\text{VAE}} + \lambda_{\text{cluster}}\, \mathcal{L}_{\text{cluster}} + \lambda_{\text{tree}}\, \mathcal{L}_{\text{tree}}, \tag{14}$$

where $\lambda_{\text{cluster}}, \lambda_{\text{tree}} \geq 0$ are loss weights. See Appendix B.3 for weighting details.

## 5 EXPERIMENTS AND RESULTS

### 5.1 SETTINGS

**Datasets.** This study uses three datasets. TCGA-LIHC provides 424 liver-cancer transcriptomic profiles with 60,661 features. TUEP (Obeid & Picone, 2016; a subset of the TUH EEG corpus) is represented by 186,200 samples with 798 features after data processing. The private PPH cohort contains 5,935 cases with 48 clinical variables. See Detailed descriptions of all datasets and data processing methods are provided in Appendix D.1.

**Performance Metrics:** Model performance is reported using accuracy, precision, recall, F1 score, and AUC; formal definition in Appendix C). Interpretability is additionally assessed via clinician responses to LLM-generated multiple-choice items.

**Experimental Setup** In this study, the encoder input dimension equals the feature count of each dataset; hidden and latent sizes are 64. Models are optimized with Adam (lr $= 10^{-3}$) for 300 epochs, using batch size 128 on PPH/TUEP and 32 on TCGA-LIHC. Results are averaged over five fixed seeds $\{19, 21, 42, 60, 99\}$. All experiments were run on 4× NVIDIA RTX 3090 (24 GB) GPUs. Code is provided in the supplementary materials.

## 5.2 EFFECTIVENESS VALIDATION OF THE INTERPRETIVE GUIDANCE MECHANISM

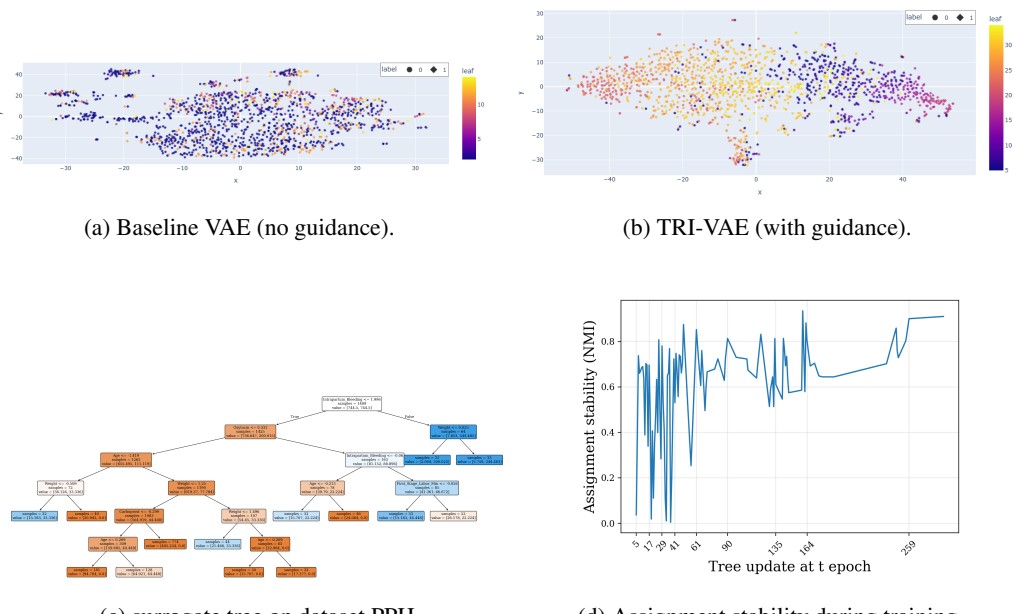

(a) Baseline VAE (no guidance).    (b) TRI-VAE (with guidance).

(c) surrogate tree on dataset PPH    (d) Assignment stability during training.

Figure 2: Panels (a–b) show t-SNE of latent space; points are colored by surrogate-tree leaf and marker shape denotes class. Panel(c) displays the surrogate tree on PPH dataset. Panel (d) reports assignment stability (NMI) across tree updates. T-SNE and surrogate-tree results for TCGA-LIHC and TUEP appear in Appendix A.1.

**Latent-space alignment.** Latent embeddings are visualized using t-SNE, as shown in Fig. 2. Without tree guidance (Fig. 2a), the embeddings are diffuse and colors corresponding to different decision paths show noticeable overlap, indicating weak alignment between latent codes and rule partitions. After applying the interpretive guidance mechanism(Fig. 2b), samples following the same decision path (denoted by similar colors) cluster more tightly in the latent space, while samples belonging to different paths are more clearly separated. This demonstrates that the latent representation becomes aligned with the rule-based partitioning induced by the decision tree, leading to a more structured and interpretable embedding space.

**Fidelity of the Surrogate Tree.** Fig. 2c shows the surrogate tree for PPH, which provides readable decision paths that account for the model's predictions. The surrogate tree closely matches the behavior of the VAE classifier on held-out data, achieving high fidelity on all datasets—PPH: 96.1%, TCGA-LIHC: 97.6%, and TUEP: 91.2%. These results indicate that the extracted paths can be trusted as faithful explanations in most cases.

**Assignment stability.** To assess the stability of tree replacement during training, the procedure tracks the NMI between consecutive leaf–assignment vectors at each tree refresh. During the first 50 epochs, the surrogate tree is freely replaced to avoid over-constraining the VAE while its representations are still forming, which manifests as early volatility in Fig. 2d. After epoch 50, the tree-update control (§4.2) is enabled to prevent dramatic structural changes that could disrupt representation learning; the curve then transitions from volatility to a stable regime, with NMI maintaining a mid–high band and gradually increasing. As the latent representation matures, updates

become conservative—mostly local refinements rather than wholesale restructurings—so consecutive leaf assignments remain stable and clusters avoid large reassignments.

Further results for TCGA-LIHC and TUEP are provided in Appendix A.1.

### 5.3 Evaluation of the Effectiveness of Tree Regularization

This experiment evaluates the tree regularization module and shows that, across datasets, it produces more compact surrogate trees and improves predictive performance. Figure 3 compares surrogate trees trained with and without tree regularization on the PPH dataset, demonstrating a clear reduction in structural complexity. AUC gains are observed — +0.059 on PPH, +0.011 on TCGA-LIHC, and +0.011 on TUEP—indicating that enforcing concise decision paths enhances generalization while also achieving higher predictive quality.

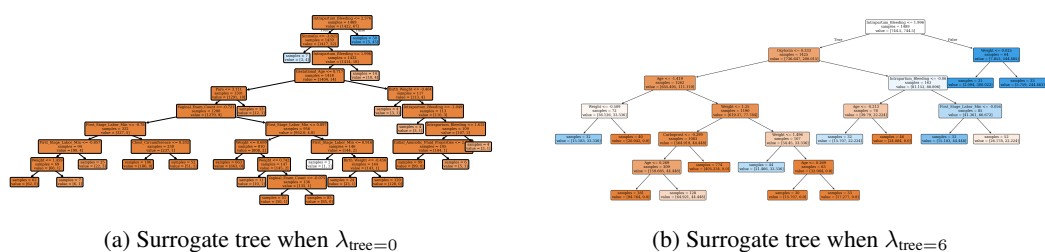

(a) Surrogate tree when $\lambda_{\text{tree}=0}$        (b) Surrogate tree when $\lambda_{\text{tree}=6}$

Figure 3: Comparison of Decision Tree Structures with/without tree regularization. See Appendix A.3 for enlarged version

### 5.4 Effect of Sampling Size in SHAP Estimation

To evaluate the efficiency of the proposed SHAP sampling strategy under high-dimensional setting, experiments were conducted under different sampling sizes on the TCGA-LIHC dataset, as shown in Table 1. Results indicate that small sampling sizes lead to only moderate performance drops but greatly reduce running time. As the sampling size increases, accuracy and fidelity improve and eventually converge when the size reaches 4000 or more. This shows that the sampling strategy maintains performance within an acceptable range while offering a clear trade-off between efficiency and accuracy in high-dimensional settings. See Appendix A.4 for the baseline comparison with Kernel SHAP.

Table 1: Performance and efficiency of sampling SHAP under different SHAP sampling sizes

| Sample Size | AUC | Accuracy | Fidelity | Time per explanation (s) |
|---|---|---|---|---|
| 10 | 0.978 | 0.918 | 0.882 | 0.231 |
| 100 | 0.985 | 0.927 | 0.906 | 2.860 |
| 1000 | 0.989 | 0.942 | 0.918 | 28.539 |
| 2000 | 0.991 | 0.952 | 0.929 | 57.187 |
| 3000 | 0.993 | 0.952 | 0.937 | 85.656 |
| 4000 | 0.995 | 0.975 | 0.965 | 114.248 |
| 5000 | 0.996 | 0.987 | 0.976 | 142.979 |

### 5.5 Explainable Prediction Ability Evaluation of TRI-VAE

TRI-VAE is evaluated on TCGA_LIHC, TUEP, and PPH datasets. All results are averaged over five fixed seeds. Baselines include a strong black-box learner (XGBoost (Chen & Guestrin, 2016), interpreted post-hoc via SHAP), an inherently interpretable generalized additive model (EBM (Lou et al., 2012; Nori et al., 2019)), VAE-based counterparts (OmiVAE (Zhang et al., 2019), XOmiVAE (Withnell et al., 2021)), and recent structurally interpretable deep models (NCART (Luo & Xu, 2024), a differentiable tree ensemble; IMN (Kadra et al., 2024), a hypernetwork yielding instance-specific linear models). Results for TCGA-LIHC, TUEP, and PPH appear in Tables 2, 3, and 4, respectively.

TRI-VAE consistently outperforms existing VAE-based models and remains competitive with other state-of-the-art interpretable models, achieving a strong balance between predictive performance and interpretability in high-dimensional data.

Table 2: comparison of model performance on dataset TCGA-LIHC

| Model | Accuracy | Precision | Recall | F1 Score | AUC |
|---|---|---|---|---|---|
| XGBoost | 0.958±0.031 | 0.987±0.014 | 0.965±0.026 | 0.976±0.018 | 0.985±0.010 |
| EBM | 0.976±0.014 | 0.987±0.009 | 0.987±0.013 | 0.987±0.008 | 0.998±0.002 |
| OmiVAE | 0.928±0.015 | 0.942±0.012 | 0.948±0.015 | 0.945±0.014 | 0.938±0.009 |
| XOmiVAE | 0.985±0.011 | 0.993±0.010 | 0.955±0.005 | 0.973±0.005 | 0.977±0.016 |
| NCART | 0.981±0.006 | **0.995±0.007** | 0.984±0.011 | **0.989±0.004** | 0.982±0.003 |
| IMN | 0.979±0.006 | 0.987±0.011 | **0.990±0.007** | 0.988±0.003 | **0.999±0.001** |
| **TRI-VAE** | **0.987±0.015** | 0.978±0.016 | 0.958±0.013 | 0.965±0.010 | 0.996±0.002 |

Table 3: comparison of model performance on dataset TUEP

| Model | Accuracy | Precision | Recall | F1 Score | AUC |
|---|---|---|---|---|---|
| XGBoost | 0.903±0.001 | 0.890±0.004 | 0.745±0.002 | 0.811±0.002 | 0.946±0.001 |
| EBM | **0.921±0.001** | 0.925±0.002 | 0.921±0.001 | 0.928±0.003 | **0.966±0.001** |
| OmiVAE | 0.844±0.002 | 0.861±0.01 | 0.844±0.003 | 0.852±0.001 | 0.809±0.001 |
| XOmiVAE | 0.897±0.003 | 0.920±0.001 | 0.958±0.001 | 0.939±0.002 | 0.934±0.001 |
| NCART | 0.890±0.007 | 0.913±0.025 | 0.959±0.033 | 0.935±0.005 | 0.926±0.007 |
| IMN | 0.845±0.003 | 0.843±0.002 | **0.999±0.002** | 0.914±0.001 | 0.865±0.001 |
| TRI-VAE | 0.906±0.011 | **0.944±0.007** | 0.940±0.013 | **0.943±0.005** | 0.936±0.005 |

Table 4: comparison of model performance on dataset PPH

| Model | Accuracy | Precision | Recall | F1 Score | AUC |
|---|---|---|---|---|---|
| XGBoost | 0.976±0.003 | 0.699±0.049 | 0.719±0.033 | 0.708±0.039 | 0.933±0.009 |
| EBM | **0.977±0.002** | 0.747±0.030 | 0.637±0.057 | 0.686±0.035 | 0.911±0.018 |
| OmiVAE | 0.825±0.014 | 0.857±0.038 | 0.807±0.048 | 0.830±0.016 | 0.829±0.020 |
| XOmiVAE | 0.923±0.021 | 0.891±0.035 | 0.915±0.028 | 0.902±0.019 | **0.938±0.015** |
| NCART | 0.976±0.004 | 0.825±0.057 | 0.558±0.041 | 0.665±0.041 | 0.935±0.012 |
| IMN | 0.975±0.004 | 0.916±0.008 | 0.531±0.06 | 0.672±0.003 | 0.934±0.001 |
| **TRI-VAE** | 0.961±0.012 | **0.967±0.013** | **0.961±0.016** | **0.963±0.011** | 0.912±0.016 |

## 5.6 EVALUATION OF EXPLAINABLE MODELS USING LLMs

In this section, explainability is assessed in the clinical context of PPH diagnosis using a locally deployed large language model (LLM) to render model outputs in a more clinician-readable form. Specifically, DeepSeek is employed, an open-source LLM that supports secure local deployment. DeepSeek was prompted with only structured outputs-namely, SHAP-derived feature rankings and symbolic, text-based representations of decision tree logic. This design intentionally avoids unconstrained generation and mitigate hallucination by grounding the LLM's generation in model-derived, domain-consistent artifacts. Prompt examples used in this evaluation are provided in Appendix B.4. DeepSeek was tasked with generating 20 multiple-choice or judgment-based questions (see Appendix B.5 for an example) strictly based on the provided artifacts. These questions were then presented to nine healthcare professionals (seven attending physicians, one associate chief physician, and one chief physician) from a local hospital. Points were awarded based on agreement between clinicians' answers and model's outputs. A weighted average score was calculated (weights: 1 for attendings, 2 for associate chief, 3 for chief physician), resulting in a final interpretability score of 86.9/100. This result supports the claim that our model's reasoning is both accessible and verifiable to domain experts, while the controlled use of the LLM ensures the reliability of the interpretability. See Appendix B.6 for a visual summary of the evaluation workflow.

## 6 CONCLUSION

This study addresses interpretability in high-dimensional modeling by introducing TRI-VAE. The method links the VAE latent space to a decision-tree structure by treating tree leaves as soft clustering labels. It further refines Kernel SHAP with a subset-size–aware sampling scheme to reduce variance and improve scalability in high dimensions. In addition, it combines intrinsic (tree-regularized) and post-hoc (SHAP-based) explanations to provide a multi-level view of model behavior. Experiments show that TRI-VAE achieves strong balance between prediction performance and interpretability, offering a practical path toward trustworthy analysis of high-dimensional data.

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

# A ADDITIONAL RESULTS

## A.1 ADDITIONAL RESULT FOR THE EVALUATION OF THE INTERPRETIVE GUIDANCE MECHANISM

### A.1.1 TCGA-LIHC

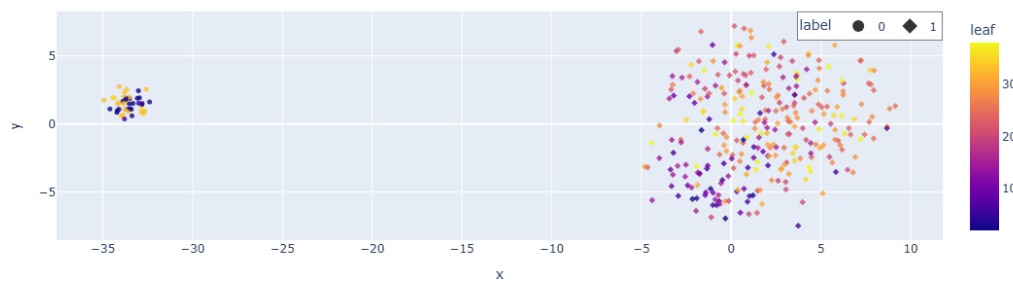

(a) Baseline VAE (no guidance).

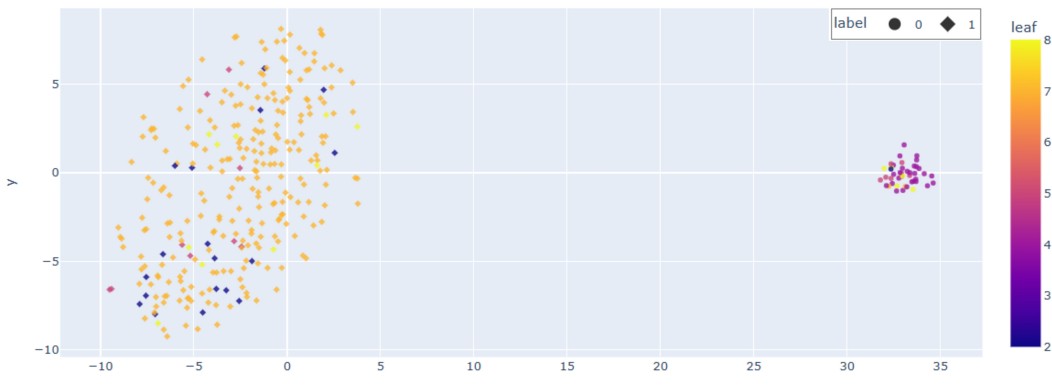

(b) TRI-VAE (with guidance).

Figure 4: TCGA-LIHC: t-SNE visualization of latent space

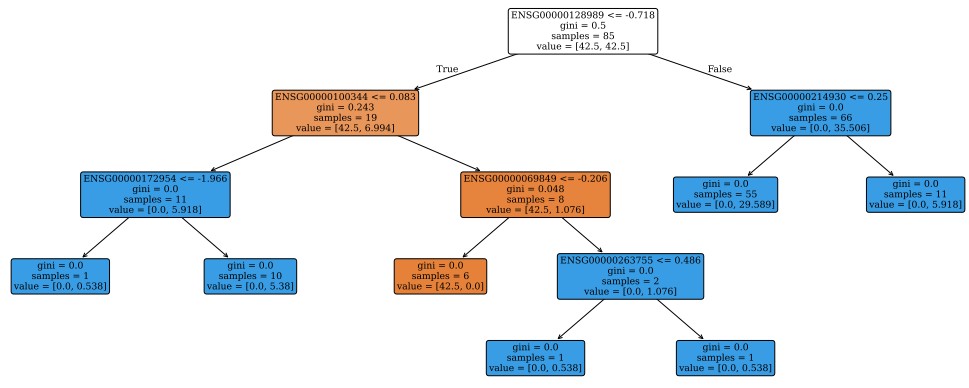

Figure 5: TCGA-LIHC: surrogate tree

## A.1.2   TUEP

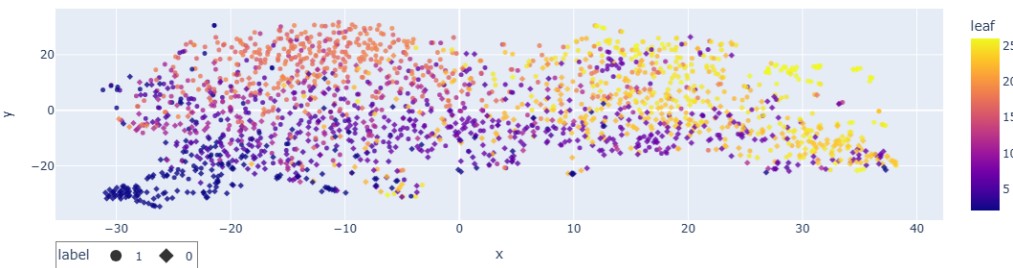

(a) Baseline VAE (no guidance).

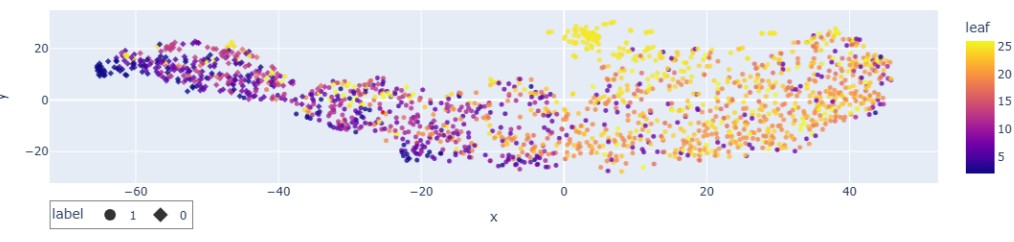

(b) TRI-VAE (with guidance).

Figure 6: TUEP: t-SNE visualization of latent space

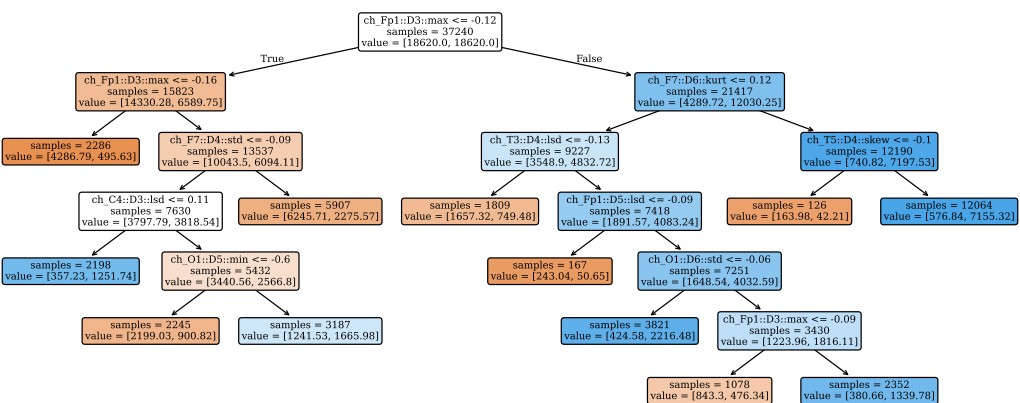

Figure 7: TUEP: surrogate tree

## A.2 EFFECTIVENESS VALIDATION OF THE INTERPRETIVE GUIDANCE MECHANISM

### A.2.1 PPH

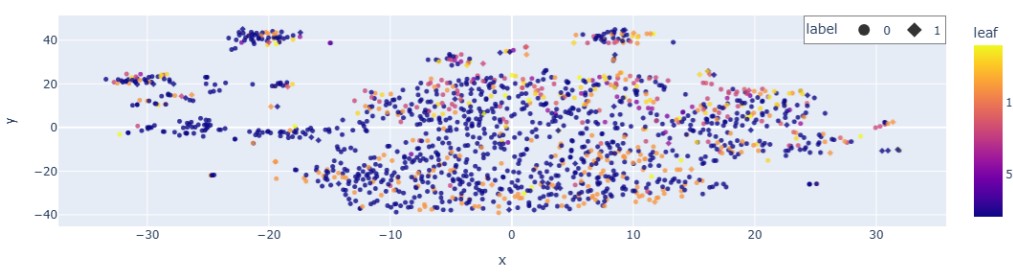

(a) Baseline VAE (no guidance).

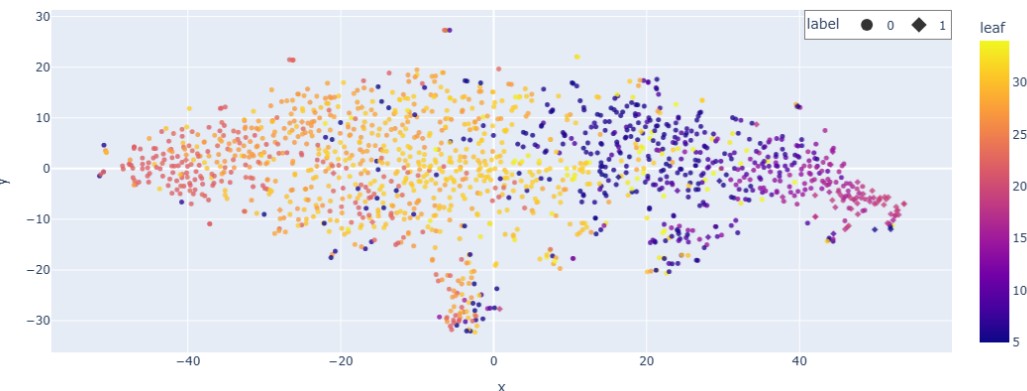

(b) TRI-VAE (with guidance).

Figure 8: PPH: t-SNE visualization of latent space

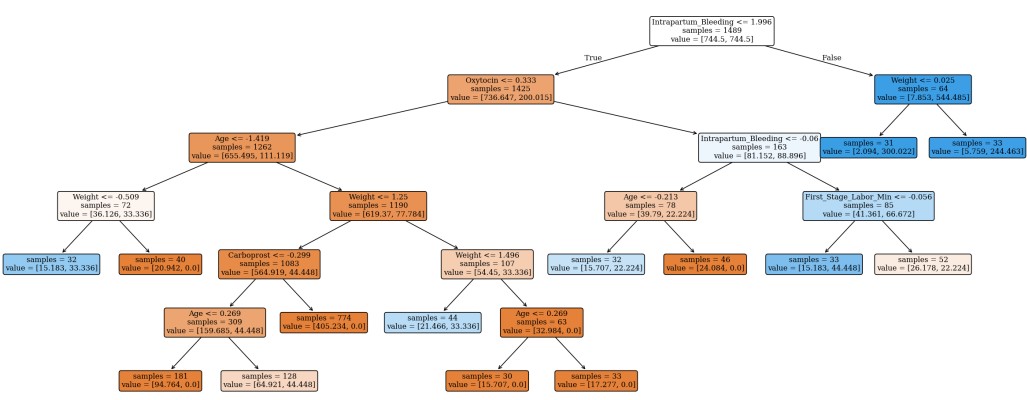

Figure 9: PPH: surrogate tree

## A.3 COMPARISON OF DECISION TREE STRUCTURES WITH/WITHOUT REGULARIZATION

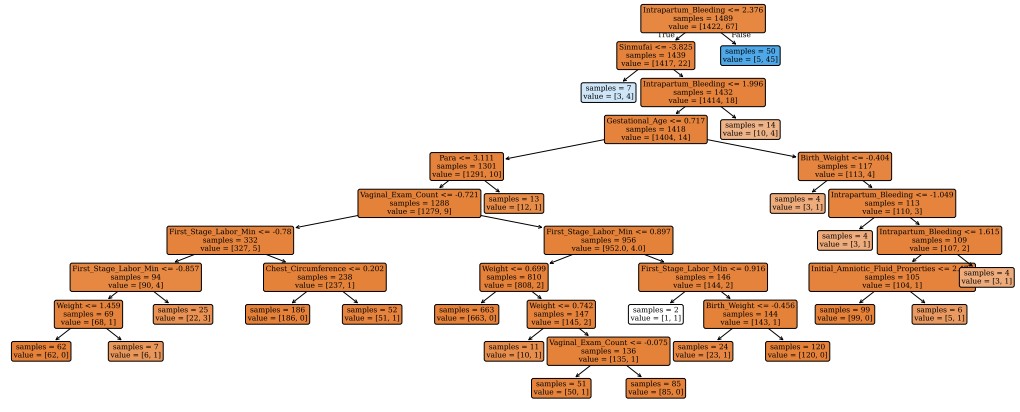

(a) Surrogate tree when $\lambda_{\text{tree}=0}$

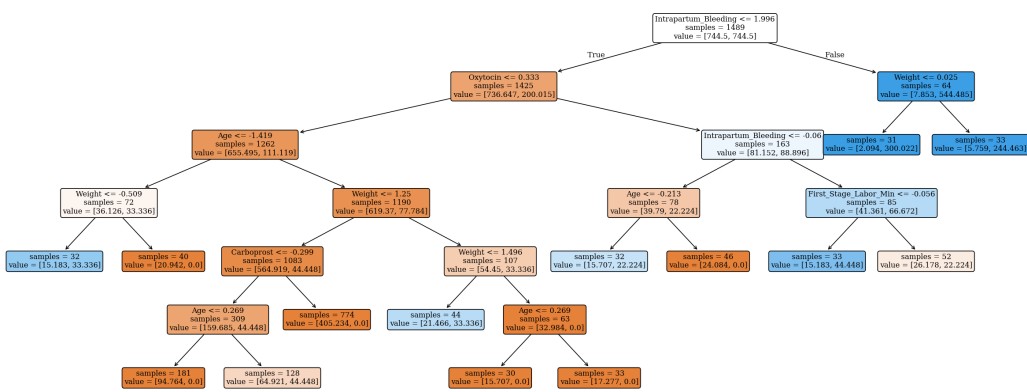

(b) Surrogate tree when $\lambda_{\text{tree}=6}$

Figure 10: Comparison of Decision Tree Structures with/without tree regularization.

## A.4 ADDITIONAL ANALYSIS ON SHAP IN HIGH-DIMENSIONAL SETTINGS

### A.4.1 KERNELSHAP BASELINE COMPARISON

Table 5: Kernel SHAP: performance and efficiency under different sampling sizes (TCGA-LIHC).

| Sample Size | AUC | Accuracy | Fidelity | Time per explanation (s) |
|---|---|---|---|---|
| 10 | 0.972 | 0.883 | 0.843 | 0.277 |
| 100 | 0.981 | 0.906 | 0.882 | 3.575 |
| 1000 | 0.986 | 0.929 | 0.902 | 35.674 |
| 2000 | 0.989 | 0.937 | 0.921 | 74.343 |
| 3000 | 0.991 | 0.941 | 0.934 | 111.353 |
| 4000 | 0.993 | 0.952 | 0.954 | 154.235 |
| 5000 | 0.994 | 0.952 | 0.954 | 193.022 |

### A.4.2 HIGH-DIMENSIONAL MEMORY/EFFICIENCY HANDLING

Directly performing Kernel SHAP in high-dimensional settings becomes memory- and time-prohibitive because it requires evaluating many coalitions and solving a large weighted least-squares problem. To make the baseline feasible and ensure a fair comparison, the same engineering measures are applied to both Kernel SHAP and the Sampling SHAP strategy:

- **Streaming mini-batch evaluation** of coalitions (no materialization of a full design matrix).

- **Mixed precision** (fp16 masks / fp32 accumulators) to reduce activation memory.

- **Conjugate-gradient (CG)** solver for the normal equations $X^\top W X \beta = X^\top W y$ instead of explicitly forming and inverting $X^\top W X$.

- **Size-stratified coalition sampling** to cover all subset sizes while controlling variance; KernelSHAP uses the Shapley kernel $w(|S|)$ with importance correction, whereas Sampling SHAP follows the proposed size distribution.

- **Batch-size capping** and seed-controlled mask generation to stabilize memory usage and variance.

These implementation choices prevent GPU memory blow-ups for Kernel SHAP in high dimension and keep both methods comparable under matched hardware/software conditions.

# B  IMPLEMENTATION DETAILS

## B.1  STRUCTURE-AWARE CLUSTER-HEAD UPDATE

---

**Algorithm 1** Structure-Aware Cluster-Head Update

---

**Require:** previous tree $T_t^{prev}$, new tree $T_t^{new}$, data $X_{U_t}$, old head $H_{\text{old}}$, initializer Init, tolerance $\varepsilon$, decay $\gamma$, IoU threshold $\tau$

**Ensure:** updated head $H_{\text{new}}$

1: **function** PATHSIM$(p_1, p_2, \varepsilon, \gamma)$
2:     $k_{\text{lcp}} \leftarrow$ length of longest common prefix under tolerant equality
       (same feature, direction, and per-split threshold difference $\leq \varepsilon$)
3:     dist $\leftarrow (|p_1| - k_{\text{lcp}}) + (|p_2| - k_{\text{lcp}})$
4:     **return** $1/(1 + \gamma \cdot \text{dist})$
5: **end function**
6: $\mathcal{P}_{\text{old}} \leftarrow$ EXTRACTPATHS$(T_t^{prev})$, $\mathcal{P}_{\text{new}} \leftarrow$ EXTRACTPATHS$(T_t^{new})$
7: $\mathbf{z}_t^{prev} \leftarrow T_t^{prev}.\text{apply}(X_{U_t})$, $\mathbf{z}_t^{new} \leftarrow T_t^{new}.\text{apply}(X_{U_t})$
8: $H_{\text{new}} \leftarrow \text{Init}(|\mathcal{P}_{\text{new}}|)$
9: **for** each new leaf $c \in \mathcal{P}_{\text{new}}$ **do**
10:     $s^\star \leftarrow \max_{o \in \mathcal{P}_{\text{old}}} \text{PATHSIM}\big(\mathcal{P}_{\text{new}}[c], \mathcal{P}_{\text{old}}[o]; \varepsilon, \gamma\big)$         ▷ best similarity value
11:     $o^\star \leftarrow \arg\max_{o \in \mathcal{P}_{\text{old}}} \text{PATHSIM}\big(\mathcal{P}_{\text{new}}[c], \mathcal{P}_{\text{old}}[o]; \varepsilon, \gamma\big)$     ▷ ID of best-matching old leaf
12:     $A \leftarrow \{i \mid \mathbf{z}_t^{prev}[i] = o^\star\}$, $B \leftarrow \{i \mid \mathbf{z}_t^{new}[i] = c\}$
13:     $\text{IoU} \leftarrow \dfrac{|A \cap B|}{|A \cup B|}$
14:     **if** $s^\star > 0 \ \wedge \ \text{IoU} \geq \tau$ **then**
15:         $w \leftarrow \sqrt{s^\star \cdot \text{IoU}}$
16:         $H_{\text{new}}[c] \leftarrow (1 - w) H_{\text{new}}[c] + w H_{\text{old}}[o^\star]$
17:     **end if**
18: **end for**
19: **return** $H_{\text{new}}$

---

## B.2 Controlled acceptance thresholds

Let

$$J_t^{\text{cur}} = \text{MSE}_{\text{val}}\big(T_t^{\text{cur}}\big) + \lambda_{\text{apl}} \cdot \text{APL}\big(T_t^{\text{cur}}; X_{\text{all}}\big),$$

$$J_t^{\text{cand}} = \text{MSE}_{\text{val}}\big(T_t^{\text{cand}}\big) + \lambda_{\text{apl}} \cdot \text{APL}\big(T_t^{\text{cand}}; X_{\text{all}}\big).$$

Here, $\lambda_{\text{apl}}$ is set to $7.5 \times 10^{-4}$. The candidate is accepted if

$$
\begin{aligned}
\text{ACCEPT}(T_t^{\text{cand}}) \iff & \big(J_t^{\text{cand}} + \varepsilon_t < J_t^{\text{cur}}\big) \\
& \wedge \big(\text{align}(T_t^{\text{cand}}, \boldsymbol{s}_t) \geq \tau_{\text{align}}\big) \\
& \wedge \big(\text{NMI}(\mathbf{z}_t^{\text{cur}}, \mathbf{z}_t^{\text{cand}}) \geq \tau_{\text{nmi}}(t)\big).
\end{aligned}
$$

Early in training, latent representations and distilled targets are nonstationary. A strict stability requirement at this stage tends to reject beneficial restructurings and underfit the surrogate. As training progresses and representations stabilize, a higher stability floor helps preserve cluster identities and prevents gratuitous structural churn. For this reason, the NMI floor is annealed from a permissive level to a stricter target.

The stability threshold is annealed over epochs ($T$ total) as

$$
\tau_{\text{nmi}}(t) = \begin{cases}
0, & t < \alpha T, \\
\tau_{\text{start}} + \dfrac{t - \alpha T}{\beta T}\big(\tau_{\text{end}} - \tau_{\text{start}}\big), & \alpha T \leq t < (\alpha + \beta)T, \\
\tau_{\text{end}}, & \text{otherwise},
\end{cases}
$$

with $\alpha = 0.25$, $\beta = 0.50$, $\tau_{\text{start}} = 0.30$, and $\tau_{\text{end}} = 0.70$ (for $T{=}300$, these correspond to $t{<}75$, $75{\leq}t{<}225$, and $t{\geq}225$). The alignment gate uses a fixed Spearman threshold $\tau_{\text{align}} = 0.35$.

### B.3 Weight Schedules for $\lambda_{\text{CLUSTER}}$, and $\lambda_{\text{TREE}}$

The training objective is a weighted composite loss:

$$L \;=\; L_{\text{VAE}} \;+\; \lambda_{\text{cluster}}\, L_{\text{cluster}} \;+\; \lambda_{\text{tree}}\, L_{\text{tree}}.$$

Let $e$ denote the current epoch, $T$ the schedule horizon (set to $T=50$ in the reported experiments), and $t = \min(1, e/T)$.

$\lambda_{\text{cluster}}$: Because soft labels derive from the surrogate's leaf structure and are particularly noisy early on, a cosine schedule (with zero initial slope at $t=0$) is adopted to avoid prematurely steering the latent representation, yet allows it to dominate later as the structure stabilizes.

The cluster-alignment weight follows a cosine rise from $0$ to $8.0$:

$$\lambda_{\text{cluster}}(t) \;=\; 8.0 \times \tfrac{1}{2}\big(1 - \cos(\pi t)\big).$$

$\lambda_{\text{tree}}$: The tree regularization weight is disabled during the first 50 epochs to avoid interfering with early representation learning, and is then fixed to a constant value thereafter:

$$\lambda_{\text{tree}}(e) \;=\; \begin{cases} 0, & e < 50, \\ 6.0, & e \geq 50. \end{cases}$$

This schedule postpones structural pressure until cluster identities and latent representation have largely stabilized; once activated, this loss term enforces compactness without introducing additional late-phase variance. The regularizer itself is given by the learned APL surrogate $\widehat{\Omega}(W;\xi)$, which maps current network parameters to an APL estimate and is trained on recent parameter–APL pairs to track model evolution.

### B.4 LLM-BASED PROMPT DESIGN FOR INTERPRETABILITY EVALUATION

To evaluate the interpretability of the TRI-VAE model from a clinical perspective, we employed a large language model (LLM) to generate a structured questionnaire targeted at medical experts. The objective is to assess whether the explanations provided by the model—based on SHAP values and decision tree rules—are intuitive and align with expert domain knowledge.

In the prompt design process, the LLM was provided with three categories of input: (1) 200 real-world anonymized clinical cases, equally split into 100 positive (PPH) and 100 negative (non-PPH) samples; (2) a ranked list of important features as identified by SHAP and decision tree analysis; and (3) the structural logic of the decision tree, including feature split thresholds and decision paths. The LLM was instructed to focus on a subset of features deemed most predictive (approximately 6 out of a total of 48), and to ignore irrelevant variables. Based on this input, the model was prompted to generate clinically meaningful questions rooted in representative patient scenarios.

---

**LLM Prompt Template**

You are a medical data analyst tasked with designing an interpretability assessment questionnaire for a clinical decision-support model used to predict postpartum hemorrhage (PPH). The model outputs are explained using:

- A decision tree comprising key features: placental width, first-stage labor duration, placental integrity, gravidity, mode of labor onset, and parity.

- SHAP value-based feature attribution highlighting: second-stage labor duration, gravidity, parity, placental width, membrane rupture type, fetal delivery mode, placental integrity, carbetocin administration, vaginal examination count, and maternal education level.

You are provided with a real-world dataset comprising 200 anonymized cases (100 PPH-positive and 100 PPH-negative patients). Your task is to generate a set of 10 representative clinical questions that will be used to evaluate the model's interpretability by practicing clinicians. Please adhere to the following instructions:

(1) All questions must be either binary (True/False) or multiple-choice.

(2) Questions derived from SHAP explanations should assess whether an increase or decrease in a specific feature value is associated with higher or lower PPH risk.
    Example: "Does greater placental width increase the risk of postpartum hemorrhage?" (True/False)

(3) Questions derived from the decision tree should be grounded in its decision paths.
    Example: "If the placental width is large, the first stage of labor is prolonged, and the patient is multiparous, is the risk of PPH elevated?" (True/False)

(4) Select approximately six representative features from the model's top-ranked features to construct clinically relevant and diverse questions.

(5) Avoid ambiguous or overly technical phrasing; the questionnaire should be understandable by medical professionals without access to model internals.

---

## B.5 Postpartum Hemorrhage Risk Questionnaire

| Question | Options |
|---|---|
| 1. Please fill out your personal information | A. Name: 
 B. Occupation: 
 C. Workplace: 
 D. Specialty/Department: |
| 2. Which of the following factors do you think is most closely related to postpartum hemorrhage risk? Please rank by importance. | A. Placental width 
 B. First stage duration 
 C. Second stage duration 
 D. Placental completion 
 E. Gravidity 
 F. Mode of delivery 
 G. Parity |
| 3. If placental width is large, the risk of postpartum hemorrhage may: | A. Increase 
 B. Decrease 
 C. Remain unchanged 
 D. Cannot determine |
| 4. Which of the following situations may lead to a higher risk of postpartum hemorrhage? (Multiple choices) | A. Lower parity 
 B. Intact placenta 
 C. Longer first stage duration 
 D. Higher level of education |
| 5. Which stage of labor (first or second stage) is more likely to increase the risk of postpartum hemorrhage? | A. First stage of labor 
 B. Second stage of labor 
 C. Cannot determine |
| 6. Does a higher parity correlate with a higher risk of postpartum hemorrhage? | A. Yes 
 B. No 
 C. Cannot be determined |
| 7. A prolonged second stage of labor may increase the risk of postpartum hemorrhage: | A. Yes 
 B. No 
 C. Cannot determine |
| 8. A higher parity is associated with a lower risk of postpartum hemorrhage: | A. Yes 
 B. No 
 C. Cannot determine |
| 9. The mode of delivery (vaginal delivery or cesarean section) does not affect the risk of postpartum hemorrhage: | A. Yes 
 B. No 
 C. Cannot determine |
| 10. There is no clear relationship between the mode of amniotic sac rupture and the risk of postpartum hemorrhage: | A. Yes 
 B. No 
 C. Cannot determine |
| 11. If the placental width is smaller and the duration of the first stage of labor is shorter, is the risk of postpartum hemorrhage lower? | A. Yes 
 B. No 
 C. Cannot determine |
| 12. If the placental width is large, the duration of the first stage of labor is long, and the parity is high, will the risk of postpartum hemorrhage significantly increase? | A. Yes 
 B. No 
 C. Cannot determine |
| 13. If the placental width is large and the placenta is intact, is the risk of postpartum hemorrhage lower? | A. Yes 
 B. No 
 C. Cannot determine |
| 14. The effect of carbetocin on postpartum hemorrhage is unrelated to other factors: | A. Yes 
 B. No 
 C. Cannot determine |
| 15. The more vaginal examinations performed, the higher the risk of postpartum hemorrhage: | A. Yes 
 B. No 
 C. Cannot determine |

| Question | Options |
|---|---|
| 16. Women with higher levels of education have a lower risk of postpartum hemorrhage: | A. Yes
B. No
C. Cannot determine |
| 17. Can postpartum hemorrhage be diagnosed in women with the following characteristics: placental width greater than 18 cm, first stage of labor duration less than 700 minutes, placental integrity score of 1 (intact), age over 30 years, placental weight greater than 500g, second stage of labor duration less than 100 minutes, and blood loss during labor less than 300 ml? | A. Yes
B. No
C. Cannot determine |
| 18. Can postpartum hemorrhage be diagnosed in women with the following characteristics: placental width greater than 20 cm, placental length greater than 20 cm, placental weight greater than 600g, first stage of labor duration less than 500 minutes, second stage of labor duration less than 100 minutes, spontaneous rupture of membranes (score 1), and age over 30 years? | A. Yes
B. No
C. Cannot determine |
| 19. Can postpartum hemorrhage be diagnosed in women with the following characteristics: placental width greater than 20 cm, first stage of labor duration greater than 800 minutes, placental integrity score of 2 (incomplete), parity of 2, second stage of labor duration greater than 150 minutes, and blood loss during labor greater than 500 ml? | A. Yes
B. No
C. Cannot determine |
| 20. Can postpartum hemorrhage be diagnosed in women with the following characteristics: first stage of labor duration greater than 600 minutes, placental width less than 17 cm, placental integrity score of 2 (incomplete), placental length greater than 20 cm, placental weight greater than 600 g, and second stage of labor duration greater than 100 minutes? | A. Yes
B. No
C. Cannot determine |
| 21. Can postpartum hemorrhage be diagnosed in women with the following characteristics: placental width greater than 18 cm, first stage of labor duration less than 700 minutes, placental integrity score of 1 (intact), placental weight greater than 500 g, second stage of labor duration less than 50 minutes, fewer than 5 vaginal examinations, and blood loss during labor less than 300 ml? | A. Yes
B. No
C. Cannot determine |

## B.6 EVALUATION PROCESS WITH LLMS

**Input:** Decision tree for key features extraction

**Input:** 100 randomly selected cases (positive:negative = 1:1)

**Prompt:** Generate questions based on real sample data and decision tree features

**Output:** Questions and options generated by LLM

**Form:** Interpretability verification questionnaire

**Action:** Consult doctors of different professional ranks

**Action:** Calculate interpretability scores by assigning weights according to professional rank

**Output:** Evaluate the model's interpretability

Figure 11: Evaluation process with LLMs

## C  EVALUATION METRICS

The following metrics are used to evaluate the performance of TRI-VAE:

**1. Accuracy**  Accuracy represents the proportion of all samples correctly classified. It is calculated as:

$$\mathrm{ACC} = \frac{TP + TN}{TP + FN + FP + TN}$$

where TP refers to the number of True Positives, TN refers to the number of True Negatives, FP refers to the number of False Positives, and FN refers to the number of False Negatives.

**2. Recall**  Recall (also known as Sensitivity or True Positive Rate) represents the proportion of actual positive samples correctly identified by the model. It is computed as:

$$\mathrm{Rec} = \frac{TP}{TP + FN}$$

**3. Precision**  Precision represents the proportion of predicted positive samples that are correctly identified. It is calculated as:

$$\mathrm{Pre} = \frac{TP}{TP + FP}$$

**4. F1 Score**  The F1 Score is a metric that combines both Precision and Recall, providing a single measure of model performance that balances the trade-off between them. It is computed as:

$$\mathrm{F1} = \frac{2 \times \mathrm{Pre} \times \mathrm{Rec}}{\mathrm{Pre} + \mathrm{Rec}}$$

This metric is especially useful when dealing with imbalanced datasets, where either Precision or Recall might be misleading on its own.

**5. AUC (Area Under the ROC Curve)**  AUC measures threshold–independent discrimination by integrating the Receiver Operating Characteristic (ROC), which plots the True Positive Rate (TPR) against the False Positive Rate (FPR) as the decision threshold varies. Formally,

$$\mathrm{AUC} = \int_0^1 \mathrm{TPR}(u)\,\mathrm{d}u \qquad \text{with } u = \mathrm{FPR},$$

**6. Fidelity (MSE).**  Given $X$ and the distilled target $y_{\mathrm{distill}}(x)$ at iteration $t$,

$$\mathrm{MSE}(T_t^{\mathrm{cand}}) = \frac{1}{|X|} \sum_{x \in X} \left( T_t^{\mathrm{cand}}(x) - y_{\mathrm{distill}}(x) \right)^2.$$

**7. Assignment stability (NMI).**  Let $z_t^{\mathrm{cur}}(x)$ and $z_t^{\mathrm{cand}}(x)$ denote leaf indices assigned by $T_t^{\mathrm{cur}}$ and $T_t^{\mathrm{cand}}$, and $P, Q$ be their empirical label distributions. Using the symmetric normalization,

$$\mathrm{NMI}(z_t^{\mathrm{cur}}, z_t^{\mathrm{cand}}) = \frac{2\,I(P,Q)}{H(P) + H(Q)} \in [0,1].$$

**8. Feature alignment (Spearman).**  Let $\mathbf{imp}(T_t^{\mathrm{cand}}) \in \mathbb{R}^{\kappa}$ denote the split–importance vector of $T_t^{\mathrm{cand}}$ restricted to $U_t$. The alignment score is

$$\mathrm{align}(T_t^{\mathrm{cand}}, \boldsymbol{s}_t) = \rho_{\mathrm{Spearman}}\big(\mathbf{imp}(T_t^{cand}),\, \boldsymbol{s}_t\big) \in [-1, 1].$$

# D   DATASETS AND PROCESSING DETAILS

## D.1   DATASETS

| Dataset | Domain | Samples | Dimensionality |
|---|---|---|---|
| TCGA_LIHC | Liver cancer, transcriptomics | 424 | 60,661 |
| TUEP | EEG for epilepsy research | 186,200 | 798 |
| PPH (private) | Postpartum hemorrhage prediction | 5,935 | 48 |

Table 7: Summary of datasets; for TUEP the figure refers to the post–preprocessing representation.

**TCGA-LIHC.**   The TCGA-LIHC dataset, provided by the University of California, Santa Cruz, is designed to support cancer research with an emphasis on liver cancer. It contains 424 samples and 60,661 features, primarily gene-expression measurements that carry transcriptomic detail relevant to liver cancer. The dataset is accessible via `https://gdc.xenahubs.net/`.

**TUEP.**   TUEP is a publicly available subset of the Temple University Hospital EEG corpus that is widely used in epilepsy research. The dataset comprises 200 subjects in total, with 100 diagnosed with epilepsy and 100 without epilepsy. In this work, each sample is represented by a 798-dimensional vector *after* preprocessing and feature extraction. The dataset is accessible via `https://isip.piconepress.com/projects/nedc/html/tuh_eeg/`.

**Private PPH.**   The private postpartum hemorrhage (PPH) dataset is sourced from a local hospital. It comprises 5,935 samples, each with 48 features, and is used for prediction of postpartum hemorrhage risk. Note that all patient data used in this process were fully anonymized prior to model access, and no personally identifiable information (PII) or protected health information (PHI) was included.

## D.2   DATA PROCESSING FOR TUEP

**Inclusion and channels.**   Recordings with labels epilepsy or no-epilepsy are included. Only average-reference montages are retained; all other montages are excluded. From each recording, 19 international 10–20 electrodes are used: Fp1, Fp2, F3, F4, C3, C4, P3, P4, O1, O2, F7, F8, T3, T4, T5, T6, Fz, Cz, Pz.

**Preprocessing.**   All EEG signals are resampled to 256 Hz. Spectral content is harmonized using a 4th-order IIR Butterworth band-pass filter with passband $0.5 - 128$ Hz. Power-line interference is attenuated with notch filters at 60 Hz and 120 Hz. No automated artifact removal is applied. Each continuous recording is partitioned into fixed, non-overlapping 10-second windows. Recordings shorter than one full window are excluded from further analysis.

**Feature Extraction.**   In this study, for each 10-s window and for every retained EEG channel, a time–frequency representation is obtained via the discrete wavelet transform (DWT). The transform employs the Daubechies–16 mother wavelet (db16) with six decomposition levels and symmetric signal extension. From the resulting decomposition, only the detail sub-bands $D_1, \ldots, D_6$ are considered, while approximation coefficients are discarded.

Let $\{a_i\}_{i=1}^N$ denote the coefficients in $D_j$ after removing any non-finite values. For each $D_j$ the following seven statistics are computed:

1. **Mean.** $\mu = \dfrac{1}{N} \sum_{i=1}^N a_i$.

2. **Population standard deviation.** $\sigma = \sqrt{\dfrac{1}{N} \sum_{i=1}^N (a_i - \mu)^2}$.

3. **Skewness.** (reported only if $N > 2$): $\text{skew}(a) = \dfrac{1}{N} \sum_{i=1}^{N} \left( \dfrac{a_i - \mu}{\sigma} \right)^3$ (bias-corrected in implementation).

4. **Kurtosis.** (non-Fisher form; Gaussian $\approx 3$; reported only if $N > 3$):

$$\text{kurt}(a) = \frac{1}{N} \sum_{i=1}^{N} \left( \frac{a_i - \mu}{\sigma} \right)^4.$$

5. **Maximum.** $\max\limits_{i} a_i$.

6. **Minimum.** $\min\limits_{i} a_i$.

7. **Log-sum distance (LSD).** $\text{LSD} = \log_{10}\left( \left| \sum_{i=1}^{N} a_i \right| + 10^{-12} \right)$, a robust energy-like scalar.

Per window, statistics from all channels and sub-bands are concatenated into a 798-dimensional feature vector:

$$\underbrace{19}_{\text{channels}} \times \underbrace{6}_{\text{sub-bands}} \times \underbrace{7}_{\text{statistics}} = 798.$$

In total, the corpus yields 186,200 windows, comprising 31,796 non-epilepsy segments and 154,404 epilepsy segments.

