# OpenReview forum: "Interpretable Variational Autoencoder with Stabilized Tree Regularization"
_ICLR.cc/2026/Conference — ICLR 2026 Conference Withdrawn Submission_

### Official Review · Reviewer_Y58M · 2025-10-31

**Soundness:** 2
**Presentation:** 2
**Contribution:** 3
**Rating:** 4
**Confidence:** 2

**Summary:**

The paper proposes TRI-VAE, a VAE coupled to a surrogate decision tree that imposes a rule-consistent partition on the latent space. The method (i) aligns latent codes to soft leaf distributions via a KL loss (“tree-guided clustering”), (ii) introduces a subset-size–aware SHAP sampler with EMA smoothing to select features robustly in high-dimensional settings, and (iii) penalizes tree complexity using a learned surrogate of Average Path Length (APL) with a stability-controlled tree-update (gating by fidelity, attribution alignment, and NMI of leaf assignments). Experiments on TCGA-LIHC (60k genes / 424 samples), TUEP EEG (≈186k / 798 features), and a private PPH cohort (≈6k / 48 features) report competitive predictive metrics, high surrogate fidelity (≈91–98%), compact trees under regularization, and an LLM-assisted clinician evaluation (weighted score 86.9/100). Code and implementation details are provided in the supplement.

**Strengths:**

1. Tree-guided soft labels + SHAP-based feature gates + APL regularization form a coherent pipeline; the controlled update and cluster-head inheritance are practical touches to stabilize training.
2. Results on omics, EEG, and a clinical tabular cohort; surrogate fidelity and t-SNEs help readers see how tree guidance shapes the latent space; tree regularization improves compactness and sometimes AUC.

**Weaknesses:**

1. The LLM-assisted MCQ evaluation measures agreement with the model’s own artifacts (tree rules, SHAP ranks), not faithfulness to ground truth clinical reasoning; n=9 clinicians with weighted scoring may overstate validity. Consider a human study that judges the factual plausibility of rules against domain knowledge or checks counterfactual consistency, independent of the model outputs.
2. Tables show TRI-VAE is competitive, but not uniformly best (e.g., EBM/IMN reach higher AUC/accuracy on TCGA-LIHC; AUC on PPH trails XGBoost). It might be good to analyze when tree-guidance helps/hurts.
3. The APL surrogate-of-a-surrogate (learn MLP to predict tree APL induced by current network) and the SHAP sampling scheme would benefit from bias/consistency discussion. It’s unclear under what conditions the APL proxy steers toward truly simpler, human-useful trees rather than gaming the estimator.
4. Soft labels are produced from a tree distilled on the model’s own predictions (and on a SHAP-selected subset), then used to train the model—risking circular reinforcement. Please clarify safeguards (e.g., separate splits for distillation vs. training, or regularized confidence).
5. Per-epoch SHAP sampling + periodic tree induction + gating can be heavy; the paper lists hardware but not wall-clock or scaling with d, κ, or sampling size beyond one table. Provide runtime/memory profiles.

**Questions:**

1. How sensitive are results to cluster and tree lambda coefficients? Is the “free-replace until 50 epochs” schedule essential? Show a sweep or a robustness table.
2. Is “fidelity” measured as accuracy of the tree on held-out predictions (teacher→student) or agreement on hard labels? Please formalize.
3. Do you have ablations for: (i) removing EMA smoothing; (ii) random vs. SHAP-selected feature subsets; (iii) disabling controlled acceptance (no gating)?

---

### Official Review · Reviewer_fFd1 · 2025-11-01

**Soundness:** 2
**Presentation:** 2
**Contribution:** 2
**Rating:** 2
**Confidence:** 4

**Summary:**

The problem tackled here is to learn an interpretable autoencoder and classifier for high-dimensional data, effectively reducing dimensionality while also gaining interpretability. The basic model is a VAE (Kingma and Welling, 2013), where the embedding is mapped to a classification head for downstream prediction. To this, they have added two mechanisms for interpretability: SHAP-based scoring for the learned embedding features (Lundberg & Lee, 2017) and tree-regularization (Wu et al 2018) that makes predictions of classification labels more "tree-like". The proposed method is thus a "Tree-Regularized Interpretable Variational Autoencoder" (TRI-VAE).

The key steps of the method are:

* only some of many features are identified as "important" for the predictions of the current tree surrogate classifier, as selected by SHAP procedure in Sec 4.3).
* at each iteration, a surrogate tree T using these *selected* features is trained
* to encourage the classifier to be "close" to the leaf-predictions of this specific surrogate tree, there's a separate "clustering head" for the classifier that maps each example to a categorical distribution over the C leaves. A standard cross-entropy loss penalizes clustering-head predictions that differ from this surrogate tree's soft labeling.
 This part is in Sec. 4.2.
* to encourage the neural net classifier to be "close" to a simple tree, a surrogate MLP is trained to map classifier parameters to an average path length, as in previous work by Wu et al.

The method is evaluated on several biomedical datasets, assessing prediction quality as well as effectiveness of tree regularization and the quality of explanations (via an LLM-derived questionaire given to human clinicians).

**Strengths:**

* I appreciate the high level goal of seeking interpretable classifiers of high-dimensional data
* Focus on biomedical datasets with some clinical use case in mind is appreciated
* Thinking about how cluster correspondence evolves over iterations is interesting and seems decently well-engineered in the latter part of Sec. 4.2
* The interest in human expert evaluation is appreciated and interesting

**Weaknesses:**

The main list of issues as I see it are here, with further elaboration in the subsections below

* T1: SHAP-selected features don't seem to impact the neural net classifier at all, just the surrogate tree
* T2: The decoder seems discarded after training: is it useful to include it in the first place?
* E1: Datasets not described sufficiently in main paper: need to know what binary classification *means* for each task, the label imbalance, and other key details.
* E2: Claims of effectiveness in 5.3 need to be verified for statistical significance
* E3: Accuracy evaluations in Sec. 5.5 needs work/explanation of uncertainty estimation
* E4: Tree and neural net evaluation are decoupled
* E5: LLM-based evaluation is interesting but lacks comparisons and key verifications
* C1: How embedding influences prediction is not clear


## Technical issues

### T1: Wouldn't it be more valuable if the SHAP-selected features were somehow prioritized in the neural nets?

I appreciate how there are SHAP criteria to select relevant features for the surrogate tree (in Sec 4.3).

But I think a weakness of the current approach is that all features (not just these) still impact the VAE and the prediction $\hat{y}(x)$. Wouldn't it be better to try to force the neural net components to avoid using the non-selected features? Otherwise I just worry about your ability to generalize well and interpretably.

### T2: Do you even need a decoder?

I don't see the decoder used in any evaluations... only the interpretability and the prediction quality seems to be assessed, not reconstructions of features. What's the purpose of building this model around a probabilistic encoder-decoder (VAE) if you'll just discard half the model as unnecessary?

To be clear, I'm not necessarily asking for decoder evaluations, unless the authors think these are important.

## Experimental issues

### E1: Dataset description needs clarity

The 3 datasets assessed are not described sufficiently in the main paper. In the main paper, I'd expect that it is clear to the reader:

* what the prediction task is (what label are we predicting? why is it useful?)
* what the features are
* what label balance/imbalance there is


### E2: Claims of effectiveness in 5.3 need to be verified for statistical significance

In Sec 5.3, the evaluation of tree regularization "effectiveness" states that gains of +0.011 AUROC are observed on two datasets (TCGA and TUEP).

Is this a meaningful gain?  I'd typically like to see significance tests to verify this.
I'd also expect that this is reported alongside specific measurements of how the APL changed before/after the intervention.
How do we know that any gains are specifically attributable to changes in APL, rather than luck in the optimization process?

### E3: Uncertainty in Sec 5.5 experiments needs work/explanation

Not clear what the "+/-" number means in the provided tables.

I feel skeptical that training big neural nets 5 separate times with different random seeds, that i can always rely on getting the accuracy result within rather narrow intervals (often the +/- is only 0.001 or 0.003 in TUEP assessment in Table 3).


### E4: Tree and neural net evaluation are decoupled

The proposed method produces a neural net and a decision tree... but it seems when the prediction quality is evaluated, the neural net is used, while when the explanations are evaluated, the tree is used. I understand why this is convenient, but this feels decoupled to me. The ultimate agreement between the classifications of the neural net and its tree is not that strong (only 91.2% on the largest dataset: TUEP). Internally, the neural net can use features the tree does not have access to.

I'd expect to see in the results tables for prediction, both the TRI-VAE neural net and the TRI-VAE-produced surrogate decision tree.



### E5: LLM-based evaluation is interesting but lacks comparisons and key verifications

There's an attempt to measure the TRI-VAE's agreement with human expert judgment, which is interesting. However, this is underdescribed in the paper and lacks any baseline value to compare to.

Apparently, the LLM came up with multiple-choice questions about mapping from inputs to predictions that were answerable by the provided surrogate tree. For example

> If the placental width is large and the placenta is intact, is the risk of postpartum hemorrhage lower?

One worry I have is that the questions here may be "easy" facts that any model (not just this one) could deduce... not clear if the 86% number reported here is impressive or not. I'd be much more convinced by a head-to-head comparison where different models need to produce MC questions, then a superset of questions is derived where all models and all clinicians need to answer, *and* clinicians need to grade the quality of the questions asked.

I also worry the hallucination potential was not verified... how do we know the answers are "correct" (in that the model's tree clearly agrees with the stated text translation providing "reasoning")?

### E6: Further experiments needed to show direct advantages compared to XOmiVAE

In terms of prediction quality, the present approach seems a bit similar to XOmiVAE (at least two datasets where neither accuracy nor AUROC differ by more than 0.02), and this approach is also methodologically similar (it also uses SHAP).

I'd expect to see a comparison about how the surrogate tree idea distinguishes the two approaches... is the average path length much better for the present approach?


## Novelty

I haven't seen a VAE-based method with the combined combination of SHAP selection and tree-regularization via average path length. Individually, as written in the paper XOmiVAE already integrates VAE with SHAP. I am not sure the present approach seems distinct enough from XOmiVAE in terms of performance (see E6), but the method difference is clear enough.

The tree regularization used here (surrogate MLP to compute average path length) seems to be taken directly from Wu et al's AAAI 2018 paper. I'll note that a citation to followup work by Wu et al (especially a journal paper on tree regularization) seems missing, but this is easily fixed in revision.


## Clarity/Presentation

### C1: In Sec 4: how does the VAE fit in to prediction of label y from high-dim. space x?

From the main paper, I do not see how the dimensionality reduction happening in the VAE influences prediction at all.
The notation of Sec 4.1-4.4 seems to map directly from x to y?

Are there some shared network modules between the encoder $q(z|x)$ and the predictor $\hat{y}(x)$? This is underdescribed in the paper and feels like a significant shortcoming at present.

### C2: In Sec 4.4: The exact loss $\mathcal{L}_{tree}$ is undefined

I'm assuming that somehow you just penalize the current network's estimated APL (via the surrogate MLP). But this needs to be mathematically crisp in the paper.


### Minor line-by-line issues with clarity

$N$ means different things in different subsections (first it is the total set of feature indices, then it is the number of data instances). Strongly advise to have each letter/symbol mean one thing throughout the paper.

Notation $p(x)$ for a distribution over leaves in Sec 4.2 seems off: if $x$ is a feature vector, the symbol $p(x)$ usually means a distribution over features, not leaves

in line 193: $\mathcal{L}_j$ denotes a leaf sequence, not a loss term like other $\mathcal{L}$ symbols

need to better define "branch orientation" in line 192-196

Eq 9: don't you need k to be an integer? You probably need to be explicit about how you round the right-hand-side

**Questions:**

I'd most like to hear from the authors about:

* RE T1: why not use SHAP-selected features to directly train the neural net classifier / encoder / decoder?

* RE T2: why train a decoder at all here? Is there a simpler approach you could have used instead?

* RE E2: is the claim of +0.011 AUROC statistically significant and practically meaningful? TCGA dataset only has <500 instances, so your test set must be pretty small... can we really be sure +0.011 is a real effect and not statistical noise?

* RE C1: can you clarify how the embedding from the VAE impacts prediction/classification?

**Details Of Ethics Concerns:**

Sec 5.6 performs an evaluation with human experts (clinicians). I don't see any description of the IRB/ethics approval for this study at present... this is key information that must be provided on submission.

---

### Official Review · Reviewer_y9Jx · 2025-11-02

**Soundness:** 3
**Presentation:** 3
**Contribution:** 2
**Rating:** 4
**Confidence:** 2

**Summary:**

This paper introduces a framework that the authors call TRI-VAE (Tree-Regularized Interpretable Variational Autoencoder). VAE is a powerful framework but lacks interpretability. Interpretability is critical in many domains including bioinformatics and healthcare.  This problem has been identified earlier and there are prior work to improve interpretability. This paper proposes a new framework along this direction.  The method integrates a variational autoencoder (VAE) for dimensionality reduction, a surrogate decision tree to impose rule-consistent structure on the latent space, and a stabilized SHAP-based attribution mechanism for feature selection and explanation consistency. The authors validate TRI-VAE across several datasets (TCGA-LIHC, TUEP, and a private PPH dataset) and introduce an LLM-assisted clinical evaluation protocol for assessing interpretability.

**Strengths:**

The paper is in a domain of high impact. The focus on biomedical datasets enhances societal relevance, and the clinician-validated LLM evaluation is an interesting contribution toward trustworthy AI. In healthcare areas  interpretability is of utmost importance. The paper validates TRI-VAE  framework with good set of experiments. It shows competitive accuracy relative to strong interpretable and black-box baselines (e.g., XGBoost, EBM, NCART, IMN), while producing concise surrogate trees and faithful explanations

**Weaknesses:**

Main weakness I find is the novelty of the approach. It appears like all the three  ingredients of the frameworks are based on prior frameworks in a straightforward manner. The paper is mostly heuristics in nature with limited theoretical clarity. Any theoretical justification or guarantees about stability, convergence, or fidelity of the learned surrogate would significantly strengthen the contribution. Having said that  , I am willing to change my opinion based on other expert reviewers input.

**Questions:**

It will be good to explicitly clarify the novelty of the framework.

---

### Official Review · Reviewer_Aawg · 2025-11-05

**Soundness:** 2
**Presentation:** 1
**Contribution:** 2
**Rating:** 4
**Confidence:** 3

**Summary:**

The paper presents TRI-VAE, which (i) learns compact representations via a VAE, (ii) aligns model behavior with a surrogate decision tree to obtain rule-based explanations, and (iii) scales SHAP-style attributions via subset-size-aware sampling. Tree updates are gated by alignment/stability criteria. Experiments use three healthcare datasets (TCGA-LIHC, TUEP, PPH).

**Strengths:**

- Combining tree-based clustering and VAE.

- Connecting to XAI metrics such as SHAP.

**Weaknesses:**

- Poor presentation. Writeup seems concatenation of different pieces and as a result, text seems not cohesive.

- The method seems merging some well-known ideas, and thus, overall novelty seems marginal.

- There are too many hyper-parameters in the overall method which makes it hard for implementation by users/practitioners.

- Numerical evaluation needs discussion of hyper-parameter selection and performing sensitivity analysis to ensure robustness.

-

**Questions:**

- List all hyper-parameters (tuning parameters) and provide data-adaptive methods to select them and further perform sensitivity analysis.

- It is unclear from Section 4.3 that how SHAP is used for subset-size–weighted sampling. The writeup needs improvements there.

- There are many other XAI metrics in the literature such as Integrated Gradient. Why SHAP is used and what happens if other metrics are used?

- What are specific benefits of using trees in this methodology?

---

### Note · Authors · 2026-01-29

I have read and agree with the venue's withdrawal policy on behalf of myself and my co-authors.

---

### Meta-Review · Area_Chair_3JRD · 2025-12-13

**Summary:**

Reviewers broadly agree that the paper combines several established ideas into a coherent pipeline targeting interpretability. The application domain and the emphasis on explainability are viewed as timely and potentially impactful. However, reviewers consistently raised concerns about limited novelty, heuristic design choices, lack of theoretical grounding, and weaknesses in experimental rigor and presentation. While some reviewers found the empirical results competitive and the pipeline reasonably well-engineered, others questioned whether the contributions go sufficiently beyond prior work.

**Reviewer Concerns:**

No rebuttal is provided.

**Reviewer Scores:**

No rebuttal is provided.

---

### Decision · Program_Chairs · 2026-01-26

Reject